# Comparative Analysis of mRNA, microRNA of Transcriptome, and Proteomics on CIK Cells Responses to GCRV and *Aeromonas hydrophila*

**DOI:** 10.3390/ijms25126438

**Published:** 2024-06-11

**Authors:** Xike Li, Yue Lin, Wenjuan Li, Yuejuan Cheng, Junling Zhang, Junqiang Qiu, Yuanshuai Fu

**Affiliations:** 1Key Laboratory of Freshwater Aquatic Genetic Resources, Ministry of Agriculture and Rural Affairs, Shanghai Ocean University, Shanghai 201306, China; 15137482558@163.com (X.L.); yuelin716@163.com (Y.L.); wjli@shou.edu.cn (W.L.); yjcheng0707@163.com (Y.C.); jlzhang@shou.edu.cn (J.Z.); 2Key Laboratory of Exploration and Utilization of Aquatic Genetic Resources, Ministry of Education, Shanghai Ocean University, Shanghai 201306, China

**Keywords:** grass carp, CIK, GCRV, *Aeromonas hydrophila*, transcriptome, proteome

## Abstract

Grass Carp Reovirus (GCRV) and *Aeromonas hydrophila* (Ah) are the causative agents of haemorrhagic disease in grass carp. This study aimed to investigate the molecular mechanisms and immune responses at the miRNA, mRNA, and protein levels in grass carp kidney cells (CIK) infected by Grass Carp Reovirus (GCRV, NV) and *Aeromonas hydrophilus* (Bacteria, NB) to gain insight into their pathogenesis. Within 48 h of infection with Grass Carp Reovirus (GCRV), 99 differentially expressed microRNA (DEMs), 2132 differentially expressed genes (DEGs), and 627 differentially expressed proteins (DEPs) were identified by sequencing; a total of 92 DEMs, 3162 DEGs, and 712 DEPs were identified within 48 h of infection with *Aeromonas hydrophila*. It is worth noting that most of the DEGs in the NV group were primarily involved in cellular processes, while most of the DEGs in the NB group were associated with metabolic pathways based on KEGG enrichment analysis. This study revealed that the mechanism of a grass carp haemorrhage caused by GCRV infection differs from that caused by the *Aeromonas hydrophila* infection. An important miRNA–mRNA–protein regulatory network was established based on comprehensive transcriptome and proteome analysis. Furthermore, 14 DEGs and 6 DEMs were randomly selected for the verification of RNA/small RNA-seq data by RT-qPCR. Our study not only contributes to the understanding of the pathogenesis of grass carp CIK cells infected with GCRV and *Aeromonas hydrophila,* but also serves as a significant reference value for other aquatic animal haemorrhagic diseases.

## 1. Introduction

Grass carp (*Ctenopharyngodon idellus*) is an important freshwater economy fish in China. It is widely popular for its delicacy, economy, and being nutrient-rich. However, due to the extremely complex and microbiologically rich environment that grass carp are exposed to during artificial intensive aquaculture [1], grass carp are highly susceptible to infection by microbial pathogens [2]. In particular, outbreaks of haemorrhagic diseases in grass carp caused by Grass Carp Reovirus (GCRV) and *Aeromonas hydrophila* can result in the death of a large number of grass carp, leading to significant economic losses and posing a serious threat to the sustainable development of China’s freshwater aquaculture industry [3,4,5].

GCRV belongs to the family Reoviridae and is composed of 11 double-stranded RNA genomic segments [6]. *Aeromonas hydrophila* is a Gram-negative bacterium that can cause potentially fatal septicaemia [3,7]. In previous studies, GCRV has been found to cause haemorrhagic diseases not only in grass carp but also in Minnow, Blackfish, and Cyprinus carpio [8,9]. *Aeromonas hydrophila* can also cause blood spots and inflammation of the gills, skin, and intestines in grass carp, croaker, sturgeon, and other aquatic animals [10,11,12].

In mammals, haematopoietic stem cells give rise to cells of the immune system [13,14]. In fish, the kidney functions as the primary haematopoietic organ, with the channel catfish kidney (CIK) being susceptible to GCRV [15,16,17,18]. CIK cells are derived from the kidney tissue of grass carp. It is a host cell line for GCRV, various viruses, and bacteria, and is a common choice for experimental studies in viral and bacterial molecular biology. The CIK cells are usually used to study inflammation [19], toxicity [20], and various other immunological analyses. These studies can elucidate the pathogenic mechanisms of viruses and bacteria, as well as the toxic effects of heavy metal ions, and provide technical support for the development and screening of novel drugs.

These studies can elucidate the pathogenic mechanisms of viruses and bacteria. Advancing omics technology is transforming biomedical research [21]. At present, mRNA and miRNA from the transcriptome and proteome have also been widely applied in the research field of aquatic animals, primarily focusing on the growth cycle [22], gonads [23,24], and immune response [3] in terms of the detection of gene expression levels in specific tissues, physiological processes, or stages, for example, Epinephelus coioides under cold stress [25] and Carassius auratus with *Aeromonas hydrophila* [26]. MicroRNAs (miRNAs) are a class of small (17–22 nucleotides), endogenous, and non-coding RNAs. They have been recognised as key regulators in various biological processes and metabolisms [27,28,29]. In animals, miRNA negatively regulates gene expression post-transcriptionally by binding complementarily to the 3′ untranslated region (UTR) of their target mRNA, leading to translational repression [30]. This interferes with the final protein output [31]. In addition, one miRNA can be the target of multiple mRNAs, and one mRNA can be regulated by many miRNAs [32]. However, the understanding of the role of miRNAs associated with grass carp disease is still limited. Isobaric tags for relative and absolute quantification (iTRAQ) are widely used in proteomics due to their numerous advantages [33,34]. These advantages include high detection efficiency, simplified mass spectrometer complexity, improved ion abundance, the enhanced coverage and reliability of protein identification, a simple and efficient labelling process, and a wide range of applications [35,36]. Proteins are the executors of life activities in animals, hence proteomic studies can better define pathogenesis. As proteomics is a newly developed technology, there are only a few reports on fish diseases, such as the gill infection of *Danio rerio* with *Aeromonas hydrophila*, the response of carp (*Cyprinus carpio*) to *Aeromonas hydrophila* post-infection, the epithelioma of spring carp virus (SVCV)-infected cells, and the VHSV-infected zebrafish cells [37,38,39,40], among others.

Although mRNA and miRNA from transcriptomics and proteomics have been utilised in the study of haemorrhagic disease in grass carp [5,41,42,43], they are typically analysed independently in most studies, resulting in a fragmented understanding rather than an integrated approach, and the pathogenesis of the disease remains unclear. Therefore, in this paper, we conducted mRNA-seq, miRNA-seq, and proteomics analyses on grass carp CIK cells infected with PBS, GCRV, and *Aeromonas hydrophila*. This is the first report on grass carp haemorrhagic disease that integrates transcriptome, miRNAome, and proteome analyses to understand the response mechanism of CIK cells following infection. In addition, KEGG and GO enrichment analyses were conducted to determine the potential functions of DEGs and DEPs. qPCR was performed to verify the reliability of RNA/small RNA-seq and iTRAQ data. These results provide important new information on the pathogenesis of CIK cell infections against GCRV and *Aeromonas hydrophila*. It further provides important new information on the immune response induced by infections of viruses and bacteria in scleractinian fishes, such as grass carp.

## 2. Results

### 2.1. Viruses and Bacteria Change Effect

CIK cells were mock-infected with PBS (20 μL, 1xPBS, N group (Control)), or infected with Aeromonas hydrophila (20 μL, 10^7^ cfu/mL, NB group) and GCRV (20 μL, 101 MOI = 0.1, NV group) at 1 × 10^6^ cells per well. No significant phenomenon was observed in the cells of the NV and NB groups after 4 h of cell infection. However, the pathogenic effects observed in CIK cells have evolved over time in response to viral and bacterial infections, exhibiting distinct effects in the two groups. In the NV group, CIK cells exhibited signs of a cytopathic effect (CPE) at 8 h, with evident CPE observed as a large number of cells merged and cavitated to form a large syncytium at 24 h. In the NB group, a small number of CIK cells had died at 8 h, while a large number of cells showed necrosis and some cells exhibited morphological changes at 24 h. These results indicate that CIK cells were infected with GCRV and *Aeromonas hydrophila* (Figure 1). The changes in the number of CIK live cells in the two groups at different time points are shown in Table 1. The results showed a gradual decrease in the number of CIK live cells over time and exhibited significant changes.

### 2.2. Preliminary Transcriptome Analysis and Identification of DEGs

Clean reads were obtained after filtering out low-quality reads, and clean bases, Q20, and Q30 are listed in the table (Appendix A). Furthermore, all reads were folded and assembled into 174,612 unigenes. The average length of the unigenes was 380 bp. The length of the unigenes sequences is mainly distributed between 200 and 500 bp (Figure 2A). In addition, the unigene sequences were annotated using online BLASTx software (https://blast.ncbi.nlm.nih.gov/Blast.cgi) (accessed on 27 November 2021) against the NCBI nr database, UniProt, and Swissprot databases. Most of the unigene sequences (65.9%) showed high homology with the model organism Danio rerio (Figure 2B). The above results show that the assembled results are credible. The genes were conserved during the evolutionary process. Principal component analysis was performed to identify the main differentiating factors based on the expression levels between the NV/N and NB/N groups (Figure 2C). There was a significant difference between N and the NB.

By comparing the gene expression levels between the N/NV, N/NB, and NV/NB groups, we identified a total of 2132 DEGs (781 up-regulated and 1351 down-regulated), 3162 DEGs (905 up-regulated and 2257 down-regulated), and 2876 DEGs (1080 up-regulated and 1796 down-regulated) (Figure 2D–G). Notably, most DEGs were down-regulated in the NV and NB groups. Comparative transcriptome analysis showed that 1458 and 2488 DEGs were screened from NV and NB groups, respectively (Figure 2H). It was interesting to note that the DEGs of NV and NB groups were significantly more numerous than the common genes shared by both groups. This shows that the pathogenesis of GCRV and *Aeromonas hydrophila* infections was somewhat different.

### 2.3. Analysis of miRNA-seq and Identification of DEMs

The sequencing of miRNA revealed the dynamic regulation of miRNA on mRNA during CIK cell infection with viruses and bacteria. Clean data were obtained after removing 3′ adaptor sequences, sequences with a base length of less than 18 nt, and filtering out rRNA, tRNA, snRNA, and snoRNA using BLAST against the ReNBase and Rfam databases (Appendix A). The length of the reads is mainly distributed in the range of 18–25 nt (Figure 3A). The above results indicate that the miRNA data can be used for further analysis. Furthermore, when the maximum genomic distance is ≤3000 bp and the number of miRNAs in each cluster is more than four, a total of 47 miRNA clusters were identified (Figure 3B). For example, the miRNA-17-92 cluster encodes miR-17-5p, miR-17-3p, miR-18a, miR-19a, miR-20a, miR-19b-1, and miR-92-1. The miRNA-17-92 cluster is mainly involved in the development of the heart, lungs, and immune system.

Compared to the N group, a total of 125 known DEMs were identified in both groups. Among them, 29 specific miRNAs and 2 novel DEMs were present in the NV group, while 26 specific miRNAs and 13 novel DEMs were identified in the NB group (Figure 3C). However, the frequency of the expression of these novel DEMs was relatively low. Furthermore, the targets of the DEMs were predicted by TargetScan and miRanda. The study identified 66 different target mRNAs for 99 DEMs in the NV group and 119 different target mRNAs for 92 DEMs in the NB group.

### 2.4. Validation of DEGs, DEMs, and Target Genes

Fourteen DEGs and six DEMs were randomly selected from the NV and NB groups. These were used to validate the reliability of mRNA-seq and miRNA-seq by RT-qPCR. The preliminary list of the selected DEGs and DEMs for RT-qPCR was provided in Appendix A. The results of RT-qPCR indicated that most of the DEGs and DEMs showed similar expression patterns to the RNA-seq or miRNA-seq data when compared with the N group (Figure 4A–H). But the expression level of the UGDH gene was opposite with RNA-Seq, and there was no significant change in the expression level of ZYX genes. This may be due to the sample homogenisation effect. The above results indicate that the mRNA and small RNA sequencing data were reliable.

To confirm the authenticity of the interactions between the miRNA and target genes, we cloned the 3′UTR regions of VDAC2 with the dre-miR-223_R+1-binding site and the 3′UTR regions of HDAC7 with the PC-3p-4801-64-binding site into the dual luciferase vector. The recombinant vector was transfected into CIK cells, and the luciferase activity was determined 24 h post-transfection. The results indicate that VDAC2 was targeted by miR-223_R+1, while HDAC7 did not exhibit a negative regulatory relationship with PC-3p-4801-64 (Figure 4I,J). This finding was not entirely consistent with the bioinformatic predictions of the target genes. This is only a reference value for further research and analysis.

### 2.5. iTRAQ Quantitative Data Analysis and Identification of DEPs

After infecting CIK cells with GCRV and *Aeromonas hydrophila*, a total of 4970 proteins were identified (Figure 5A). The mass error of the protein was controlled to be less than 0.05 Da. This result indicates that the error distribution was primarily around 20 (Figure 5B). In addition, the length of the peptides was mainly distributed between 7 and 20 amino acids (Figure 5C). The most identified proteins were mainly composed of 1–4 peptides or more than 11 peptides (Figure 5D). Approximately 50% of the identified peptides had a protein coverage of more than 10% (Figure 5E). The above results indicate that the quality of the iTRAQ data aligns with the subsequent analysis and research characteristics. Moreover, compared with the N group, we identified 627 differentially expressed proteins (DEPs) in the NB group, with 291 up-regulated and 336 down-regulated, as well as 712 DEPs, with 416 up-regulated and 296 down-regulated. A comparative proteome analysis also revealed that 385 specific proteins (181 up-regulated and 204 down-regulated) were identified in the NV group, while 470 specific proteins (304 up-regulated and 166 down-regulated) were identified in the NB group from the DEPs (Figure 5F).

### 2.6. Functional Annotation Based on GO and KEGG Analysis

GO enrichment analysis was performed on the DEGs/DEPs to verify their potential functions. This provided the GO functional annotation for DEGs/DEPs of NV and NB groups (Appendix A). In the NV group, the DEGs related to Biological Processes (BP) were Cellular Processes, Single Organism Processes, and Metabolic Processes. In the Cell Component (CC) category, mainly the cell, cell part, and organelle terms were enriched, while in the Molecular Function (MF) category, the binding and catalytic activity terms were mainly enriched. The GO functional annotation of the DEGs/DEPs in the NB group was similar to that of the DEPs in the NV group. Additionally, it was similar to the GO functional annotation of the DEGs in the NV group. Most GO terms in both groups belonged to the biological process category. This indicates that a series of molecular events occurred in grass carp after infection. In addition, the top 30 GO terms associated with specific genes in the NV and NB groups are also presented (Appendix A). In the NV group, most of these terms are associated with the nucleus, integral membrane, and cytoplasm. In the NB group, almost all of these terms are associated with ATP binding and the cytoplasm. The GO enrichment results of 385 and 470 specific proteins in the NV and NB groups were consistent with the GO annotation of the DEGs (Appendix A).

KEGG was helpful for further understanding the bioinformatics pathways of DEGs/DEMs/DEPs for molecular interactions in CIK cells. The pathway enrichment analysis of the targets of the DEMs is presented in Table 2. The metabolism and the immune system were the most frequently represented pathways in this analysis. This result indicates that DEMs are related to the immune response, growth, and development of CIK cells during GCRV and *Aeromonas hydrophila* infection. In the NV group, 831 DEGs with annotations were mapped to 224 pathways, while in the NB group, 1089 DEGs with annotations were mapped to 230 pathways. Some genes were unannotated in GO and KEGG, possibly representing genes with unknown functions or newly discovered genes. The KEGG enrichment analysis of DEGs in the NV and NB groups revealed that they were primarily enriched in signalling, cell proliferation and death, endocytosis, immunity, and disease infection categories (Appendix A). This result indicates that grass carp responded strongly to both GCRV and *Aeromonas hydrophila* infection. In addition, some pathways related to human diseases were enriched by DEGs. This suggests that the function of these pathways may be related to cell growth, development, and vitality. The results of the KEGG pathway annotation revealed that the DEPs of the NV group were mainly enriched in the ribosome and calcium signalling pathway, while the DEPs of the NB group were mainly enriched in the ribosome and PPAR signalling pathway. The top 20 most enriched pathways are listed (Appendix A). In addition, most pathways of the NB group were enriched in metabolic pathways, while most pathways of the NV group were enriched in environmental information processing.

It was interesting to note that specific genes of the NV group were mainly involved in molecular process pathways, such as focal adhesion and endocytosis. Whereas specific genes of the NB group were mainly related to metabolic pathways, such as the purine metabolism and pyrimidine metabolism. Large specific proteins of the NV group were significantly enriched in endocytosis and the MAPK signalling pathway. The second largest specific proteins were associated with transport and catabolic pathways, such as endocytosis and phagocytosis. However, the specific proteins of the NB group were related to cell migration, such as the regulation of the actin cytoskeleton and focal adhesion, as well as metabolic pathways like the amino acid metabolism and fatty acid metabolism (Appendix A). The above results show that the majority of the DEPs were frequently involved in cellular processes following viral infection, while most of the DEPs were extensively engaged in metabolic pathways after bacterial infection. This result was similar to the pathway enrichment of specific genes. This suggests that the infection mechanism of GCRV is different from that of *Aeromonas hydrophila*. In addition, immune responses were essential in both groups, and the immune pathways influenced by the DEGs and DEPs in the NV and NB groups were listed (Appendix A).

After infection with GCRV and *Aeromonas hydrophila*, most of the DEPs were involved in the ribosome pathway. In addition, the GO terms related to transport and translation were enriched for a large number of DEPs. The above results indicate that the protein translation process was impaired after CIK cell infection. There were 18 (6 up-regulated, 12 down-regulated) DEPs in the NV group and 23 (9 up-regulated, 14 down-regulated) DEPs in the NB group in the ribosome pathway (Appendix A).

### 2.7. Correlation Analysis of DEM, DEG, and DEP

All identified reliable proteins and the transcripts of their corresponding genes were compared and analysed for their respective expression changes in accordance with the quantitative information (Figure 6A,B). Genes and proteins in the third and seventh quadrants showed significant differences. The expression patterns were consistent. Genes and proteins in quadrants 1 and 4 showed lower protein expression abundance compared to mRNA. Conversely, genes and proteins in quadrants 6 and 9 exhibited higher protein expression abundance than mRNA. Furthermore, hierarchical clustering was performed to visually demonstrate the changes in the expression levels of common DEGs and DEPs in the N, NV, and NB groups (Figure 6C). This suggests that their expression patterns in the transcriptome and proteome were different. These genes, which were expressed at opposite protein levels, may undergo post-transcriptional degradation or over-expression due to regulatory factors.

To further understand the pathogenesis of CIK cells after GCRV and *Aeromonas hydrophila* infection, an interaction network was constructed by integrating DEMs, DEGs, and DEPs constructed on the NV group (Figure 7) and on the NB group (Figure 8). The network indicated that miRNAs, mRNAs, and proteins have potential regulatory relationships. In general, miRNAs negatively regulate the expression of their target genes. Therefore, there were eight negative miRNA–mRNA–protein interactions in the NV group, involving 27 DEMs, 8 DEGs, and 1 DEP. These genes are mainly related to endocytosis and the immune system. For example, comp322729_c0 (the inhibitor of nuclear factor kappa-B kinase gamma subunit, Ikbkg) is involved in multiple immune pathways (ko04620, ko04621, ko04622, ko04623, ko04660, and ko04662). This implies that Ikbkg plays an important role in the process of host and virus antagonism during infection. Meanwhile, dre-miR-24-5-p5 negatively regulated the expression of Ikbkg. There were 31 negative miRNA–mRNA–protein interactions in the NB group, involving 49 DEMs, 31 DEGs, and 27 DEPs. These genes were mainly related to the purine metabolism, tight junctions, and the immune system. For example, comp67711_c0 (classical protein kinase C, Prkca) was annotated in the immune pathways (ko04650, ko04666, and ko04670). When involved in the natural killer cell-mediated cytotoxicity pathway (ko04650), it is associated with the resistance mechanism against early bacterial infections. Meanwhile, dre-miR-31 negatively regulated the expression of Ikbkg.

## 3. Discussion

The demand for and quality of aquatic products have gradually increased with the improvement of people’s living standards. In addition, the decline of natural fishery resources has also promoted the development of the aquaculture industry [44]. Aquatic animals live in a complex environment abundant in various microorganisms [45,46]. Among these, haemorrhagic diseases caused by GCRV and *Aeromonas hydrophila* are likely the most prevalent in aquaculture [47,48]. These diseases can result in a mortality rate of up to 85%, making them a primary cause of economic losses in grass carp farming [5]. Therefore, in the current study, we analysed the transcriptome of miRNA, mRNA, and proteome in CIK cells infected with GCRV and *Aeromonas hydrophila* to gain insight into the pathogenesis of haemorrhagic disease in grass carp. DEMs, DEGs, and DEPs were screened from miRNA, mRNA transcripts, and proteins to identify their potential functions. This study not only improved the profound understanding of grass carp, but also provided valuable insights for the treatment and prevention of haemorrhagic diseases in other aquatic animals.

In the current study, 2132 and 3162 DEGs were identified in the NV and NB groups, respectively. There were more down-regulated DEGs than up-regulated DEGs. This result was similar to the identification of *Aeromonas hydrophila* in grass carp intestines [49]. It is difficult to determine the exact number of DEGs. This is because the environment of cell infection is different. On the whole, the results of GO annotation of DEGs and specific genes in both groups were consistent with those of grass carp infected with *Aeromonas hydrophila* [49] and Silver Crucian Carp infected with GCRV [50]. GO enrichment analysis showed that BP mainly contained terms related to the cytoplasm, cellular processes, and single organism processes. In the Cellular Component (CC) category, terms mainly included cells and cell parts. Terms related to binding and catalytic activity were mapped to MF. There were similar annotation results in Hu’s study [17]. The cytoplasm is an essential site for protein primary structure formation and is involved in various biochemical reactions. It plays a crucial role in the extensive exchange of information between the nuclear and cytoplasmic compartments, thus maintaining cellular homeostasis. [51], as well as in the function of exosomes in the cytoplasm [52]. ATP binding can alter the constitutive activity of the membrane channel. It will cause the disruption of the internal and external transport of many substances when the binding ability is altered [53]. It is worth noting that the Focal Adhesion and Endocytosis pathways exhibited the significant enrichment of DEGs during virus infection. This finding is similar to the results observed in Grass Carp CIK cells infected with GCRV for 8 to 24 h. Most DEGs were also concentrated in the endocytosis and transport pathways [8]. During bacterial infection, DEGs were primarily concentrated in the metabolic pathways [12]. This suggests that the infection mechanism of GCRV is different from that of *Aeromonas hydrophila*. Endocytosis regulates the exchange of substances between cells and the surrounding environment through cell deformation [54]. It plays an important role in maintaining the dynamic equilibrium and cell signalling of the cell [44]. These results showed that GCRV and *Aeromonas hydrophila* had significant physiological and biochemical effects on CIK cells.

The miRNA plays an important role in elucidating the regulation of biological functions [55]. To date, several studies have been published on the evaluation of the role of miRNAs in viral or bacterial infections [56,57]. In this study, we identified 99 known DEMs in the NV group and 92 known DEMs in the NB group. All DEMs were down-regulated in the NV group, while 14 DEMs were up-regulated and 78 DEMs were down-regulated in the NB group. The microRNAs such as miRNA-1, miRNA-133, and miRNA-499-5p showed higher expression in the NB group. It has been reported that miR-499-5p can promote the formation of oxidative muscle fibres by downregulating the expression of the pSox6 gene in pigs [58]. The miRNA-1 and miRNA-133 can not only regulate skeletal muscle development in ducks [59], but also modulate the mesodermal development of pluripotent stem cells, which plays an important role in heart development and physiopathology [60]. The miRNA-1 also participates in inhibiting the migration and invasion processes of the *HEp2* laryngeal squamous carcinoma cell line [61]. These findings suggest that these miRNAs are associated with cell motility following viral and bacterial infections, which may account for the changes in CIK cell morphology observed in this study. The targets of DEMs were predicted using TargetScan and miRanda to enhance the accuracy of target gene prediction. The 66 and 119 target mRNAs and 99 and 93 DEMs in the NV and NB groups, respectively, were subjected to functional annotation. This result shows that most of the DEMs’ target genes were frequently enriched in metabolic and immune pathways during GCRV and *Aeromonas hydrophila* infection. This result was similar for Grass Carp in response to GCRV infection [62], Crucian Carp infected by *Aeromonas hydrophila* [63], and Blunt Snout Bream infected by *Aeromonas hydrophila* [64] in antigen processing and presentation, the Toll-like receptor signalling pathway, NOD-like receptor signalling pathway, chemokine signalling pathway, and B cell receptor signalling pathway. Toll-like receptors sense pathogen invasion and ultimately eradicate the invading microbes [65], particularly in inflammatory diseases [66]. NOD-like receptors are crucial inflammatory regulators [67]. Chemokines are potential regulators of cell migration in continuous immune surveillance, inflammation, homeostasis, and development [68]. T cells and B cells are vital effectors of the adaptive immune response and pathogen elimination [69,70].

Ribosomes are molecular machines for protein synthesis. They play a crucial role in protein translation [71]. Ribosomes are composed of RNA and proteins [72], and the assembly of eukaryotic ribosomes requires many proteins to participate [73]. In total, 18 DEPs (6 up-regulated, 12 down-regulated) in the NV group and 23 DEPs (9 up-regulated, 14 down-regulated) in the NB group were found to be enriched in the ribosome pathway based on the KEGG annotation of the DEPs in the respective groups. The results indicated that the ribosome pathway was significantly affected by CIK cells in response to GCRV and *Aeromonas hydrophila* [74]. The *eIF-3* was also detected in the differential proteins. It played an important role in the formation of the initial translation complexes and in improving the translation accuracy [75], inconsistent with the annotation results of differentially expressed acetylated proteins [76]. After GCRV and *Aeromonas hydrophila* infection, stress granules (SGs) are formed containing mRNA to initiate translation inhibition and affect protein expression [42]. The actin cytoskeleton was found to be associated with cell migration and morphology [77]. The majority of the differential proteins were primarily enriched in the regulation of the actin cytoskeleton pathway. This result suggests that CIK cell fusion and migration were significantly affected in response to GCRV and *Aeromonas hydrophila* infection. For example, the β1 integrin was able to support the polymerisation of fibronectin fibrils and promote cell adhesion in GD25 cells [78,79]. In addition, the DEPs of both groups indicated that the interaction between CIK cells and the pathogen elicits a robust immune response. A similar phenomenon occurred in teleosts [80,81]. Innate immunity serves as the first line of defense against external stimuli [45], while T cells and B cells are the key effector cells of adaptive immunity [82,83].

A total of 14 DEGs and 6 DEMs were randomly selected from the NV and NB groups. The reliability of the mRNA-seq/small RNA-seq data was validated by RT-qPCR. The results of the RT-qPCR showed that most miRNA/mRNA exhibited similar expression trends to the mRNA-seq/small RNA-seq data. However, there was no significant change in the expression level of the down-regulated specific genes ZYX and UGDH in the NV group, which showed up-regulated expression tendencies. This discrepancy may be attributed to the sample homogenisation process [44].

DEPs and their corresponding DEGs with their respective expression changes were compared and analysed based on the quantitative information. Some genes and proteins exhibited opposite trends, which could be due to post-transcriptional degradation or overexpression influenced by regulatory factors [84] and the integrated transcriptome of mRNA, miRNA, and the proteome. An interaction network was constructed by integrating DEMs, DEGs, and DEPs. There were eight negative miRNA–mRNA–protein interactions in the NV group, involving 27 DEMs, 8 DEGs, and 1 DEPs. These genes are mainly related to endocytosis and the immune system. For example, the mutation of the *Ikbkg* gene can cause an immune deficiency in humans [85]. In the NB group, there were 31 negative miRNA–mRNA–protein interactions involving 49 DEMs, 31 DEGs, and 27 DEPs. These genes are mainly related to the purine metabolism, tight junctions, and the immune system. Prkca was resistant to the induction of Th17 cell-dependent experimental autoimmune encephalomyelitis in vivo [86]. In summary, the above results have shown that both the innate and adaptive immune systems are highly responsive to viral and bacterial infections of CIK cells.

## 4. Materials and Methods

### 4.1. Cells’ Infection and Sample Collection

Grass carp kidney cells (CIK) were cultured in monolayer of grass carp kidney tissue and cell lines were established in January 1982 by Yang Xianle’s group [87]. *C. idellus* (*Ctenopharyngodon idellus*) kidney (CIK) cells, *Aeromonas hydrophila* (AH10), and Grass carp reovirus (GCRV, JX01) were generously provided by Yang Xianle’s research team at the National Aquatic Pathogens Repository, Shanghai Ocean University. CIK cells were plated in six-well plates and cultured in Medium 199 supplemented with 10% fetal bovine serum (FBS, Gibco, Carlsbad, CA, USA) and antibiotics (100 mg/mL penicillin and streptomycin) at 28 °C. CIK cells were mock-infected with PBS, or infected with *Aeromonas hydrophila* and GCRV at 1 × 10^6^ cells per well. Samples were collected at 0, 4, 8, and 24 h post-infection (HAI). Control and experimental samples were mixed separately for RNA and protein extraction. The samples were immediately stored at −80 °C and used for further analysis. These harvested samples were used for performing mRNA-seq, miRNA-seq, and proteomics analyses. Three biological replicates were conducted for each treated group at each time point.

### 4.2. RNA Isolation, cDNA Library Construction, and Transcriptome Sequencing

Total RNAs were extracted using Trizol reagent (Invitrogen, Carlsbad, CA, USA) according to the manufacturer’s instructions. The quantity and quality of the RNA were assessed using an Agilent 2100 Bioanalyser and 1% agarose gel electrophoresis. The purity of the RNAs was assessed by determining the absorbance at 260/280 (greater than 1.8 and less than 2) and 260/230 using the RNA 6000 Nano LabChip Kit (Agilent, Santa Clara, CA, USA).

The RNA samples from the three groups at 8 h after infection (HAI) were combined and utilised for mRNA-Seq and miRNA-Seq analysis. For the preparation of cDNA libraries, cDNA was ligated to the adapter, and fragments of 300 bp were purified using agarose gel electrophoresis. The mRNA-seq was performed using the Illumina HiSeq2500 at LC-BIO in Hangzhou, China. A small RNA library was prepared following the protocol of the TruSeq Small RNA Sample Prep Kits (Illumina, San Diego, CA, USA) using approximately 1 µg of total RNA. Single-end sequencing (36 bp) was then performed on an Illumina HiSeq2500 at LC-BIO (Hangzhou, China) according to the manufacturer’s recommended protocol. Notably, the small RNA sequencing was independent of the mRNA transcriptome sequencing.

### 4.3. Preliminary Analysis of RNA-seq Data

Clean data were obtained after removing low-quality reads, 5′ and 3′ adaptor reads, and reads where the proportion of N is greater than 5% of the total reads. The TopHat2.1.1 software [88] was used to map the clean data to the reference genome of the grass carp [89]. The HTSeq2.0.3 software was used to count the number of reads mapped to each gene [90]. The results for Q20, Q30, and GC are shown in Table 1.

The expression levels of the unigenes were calculated under various treatment conditions. First, the Salmon and Kallisto transcripts Per Million (TPM) method was used to calculate and normalise the expression levels of the unigenes [91]. Furthermore, a differential expression analysis between NV, NB, and N groups was conducted using the DESeq package [92] when genes with an adjusted *p*-value ≤ 0.05 and |fold change| ≥ 2 in the DESeq analysis were identified as DEGs. If a DEG existed only in the NV or NB group, it was identified as an NV- or NB-specific differential expression gene (DEG).

### 4.4. Data Analysis and Bioinformatics Analysis of Small RNA

The raw reads were processed to remove adapter dimers, junk, low complexity, common RNA families (rRNA, tRNA, snRNA, and snoRNA), and repeats by blasting against the GenBank database (http://blast.ncbi.nlm.nih.gov)(accessed on 19 December 2020) and the Rfam database (http://sanger.ac.uk/software/Rfam) (accessed on 19 December 2020). Clean small RNA reads were obtained. Subsequently, unique sequences 18 to 26 nucleotides in length were mapped to specific species precursors in miRBase 20.0 through a BLAST search to identify known miRNAs and novel 3p- and 5p-derived miRNAs. One base mismatch is allowed during the alignment process, and the sequences mapped to specific species with mature miRNAs were identified as known miRNAs. Finally, the unmapped sequences were further BLASTed against specific genomes, and the hairpin RNA structures containing sequences were predicted from the flanking 80 nt sequences using the RNAfold software (http://rna.tbi.univie.ac.at/cgi-bin/RNAfold.cgi) (accessed on 19 January 2021). The expression of miRNA in different groups was normalised by transcripts per million (TPM). The significance threshold was set at |log2| ≥ 2 and *p*-value ≤ 0.001 for each test.

### 4.5. Validation of mRNA and miRNA by RT-qPCR

To validate the reliability of mRNA-seq and small RNA-seq, 14 DEGs and 6 DEMs were randomly selected for RT-qPCR. The specific primers for quantitative analysis were designed using Primer 5.0 software. The mRNA cDNA synthesis was performed using the PrimeScript real-time PCR kit (Takara, Dalian, China) following the manufacturer’s protocols. The relative expression of six mature miRNAs was selected for analysis using miRNA stem-loop quantification. Reverse transcription and forward primers for miRNAs were designed using miRNA Design V1.01 software, while the reverse primers were universal primers. The first strand of miRNA cDNA synthesis was synthesised using the stem-loop method with the miRNA 1st Strand cDNA Synthesis reverse transcription kit (Vazyme MR101-01, Nanjing, China). RT-qPCR amplification reactions were performed using the Power SYBR Green PCR Master Mix Kit (Applied Biosystems, Carlsbad, CA, USA) for mRNA and the miRcute Plus miRNA qPCR Detection Kit (Tiangen, Beijing, China) for miRNA on an ABI ViiA™7 instrument (ABI, Carlsbad, CA, USA). The reaction system for both mRNA and miRNA was 20 μL each. GAPDH and miR-22a were used as internal controls for mRNA and miRNA, respectively. Three biological replicates were independently measured for RT-qPCR. The relative expression level of each gene was evaluated using the 2−ΔΔCT method. Results are expressed as mean ± standard deviation.

### 4.6. Target Prediction and miRNA Overexpression

To predict miRNAs targets, two computational target prediction algorithms (TargetScan 50 and miRanda 3.3a) were used to identify miRNA-binding sites according to the complementary region between miRNAs and mRNAs and the thermodynamic stability of the miRNA-mRNA duplex. Finally, the data predicted by both algorithms were combined and the overlap was calculated. The 3′UTR sequences of HDAC7 and VDAC2 and the binding sites of PC-3p-4810-64 and dre-miR-223_R+1 were cloned into the pmir-GLO vector. CIK cells in 24-well plates were transfected with 100 nM PC-3p-4810-64 and dre-miR-223_R+1 mimics or mimics control (GenePharma). Lipofectamine 3000 (Invitrogen, Karlsruhe, Germany) was used for transfection. Dual-luciferase activity was detected at 24 h post-transfection using a dual-luciferase reporter assay system (Promega) according to the manufacturer’s instructions.

### 4.7. Protein Sample Preparation, iTRAQ Labelling, and LC-MS/MS Analysis

Collected CIK cell samples were ground to powder in liquid nitrogen and extracted with the lysis buffer (7 M urea, 2 M thiourea, 4% CHAPS, 40 mM Tris-HCl, 1 mM PMSF, 2 mM EDTA). Subsequently, 10 M DTT was added to the sample, which was then sonicated three times at 200 W for 15 min. Finally, total proteins were extracted after centrifugation at 4 °C, 30,000× *g* for 15 min. Protein concentrations were determined using the Bradford method as previously described [93]. The proteins in the supernatant were stored at −80 °C for further analysis.

In total, 100 µg of protein was collected from each group and digested with trypsin gold (Promega, Madison, WI, USA) at a protein:trypsin ratio of 30:1 at 37 °C for 16 h, as previously described [94]. Then, samples were labelled with the iTRAQ tags as follows: N group with 116 tags, NV group with 121 tags, and NB group with 119 tags. The labelled peptide mixtures were then pooled and SCX chromatography was performed using an LC-20AB HPLC Pump system (Shimadzu, Kyoto, Japan).

The samples were pre-fractionated by SCX and then analysed by LC-ESI-MS/MS. Data acquisition was performed on a TripleTOF 5600 System (AB SCIEX, Concord, ON, Canada) equipped with a Nanospray III source (AB SCIEX, Concord, ON, Canada). The data were acquired by setting the following parameters: ion spray voltage of 2.5 kV, curtain gas of 30 psi, and nebuliser gas of 15 psi. The mass spectrometer was operated with a resolution of ≥30,000 for time-of-flight (TOF) mass spectrometry scans. The total cycle time was fixed at 3.3 s, and four time bins were summed for each scan at a pulse frequency of 11 kHz by monitoring the 40 GHz multichannel TDC detector with four anode channel detections. Dynamic exclusion was set for half of the peak width (5 s).

### 4.8. Bioinformatics Analysis of iTRAQ Data

Raw data files from the Orbitrap were converted to MGF files using Proteome Discoverer 1.2 (PD 1.2, Thermo, Carlsbad, CA, USA). Protein identification was performed using the Mascot search engine (Matrix Science, London, UK; version 2.3.02) against the database “C_idella_female_genemodels.V1.aa.gz (32,926 s)” (http://www.ncgr.ac.cn/grasscarp/) (accessed on 18 October 2021). Meanwhile, to reduce the probability of false peptide identification, only peptides with significance scores (≥20) at the 99% confidence interval by a Mascot probability analysis greater than “identity” were counted as identified. The quantitative protein ratios were weighted and normalised by the median ratio in Mascot. We only considered values with a *p*-value ≤ 0.05, and fold changes |log2| ≥ 1.2 were designated as DEPs. If a DEP was present exclusively in either the NV group or the NB group, it was classified as a specific DEP of the NV group or the NB group, using KEGG and GO Pathway Enrichment Analysis.

Functional annotation of genes and proteins was performed using the Blast2GO program and online BLASTx software (https://blast.ncbi.nlm.nih.gov/Blast.cgi) (accessed on 27 November 2021) against the non-redundant protein database (NR; NCBI), UniProt, and SwissProt databases. Additionally, Kyoto Encyclopedia of Genes and Genomes (KEGG) and Gene Ontology (GO) enrichment analyses were carried out to ascertain the potential biological functions of DEMs’ target genes, DEGs, and DEPs. GO function classification and visualisation of the annotation results are based on WEGO (Web Gene Ontology Annotation Plot, v2.0) software [95]. The DAVID bioinformatics resource version 6.7 was utilised for enriching the KEGG pathway analysis of genes and proteins [96]. KEGG and GO enrichment analysis of DEGs and DEPs with a *p*-value < 0.05 were considered statistically significant.

### 4.9. Integrated mRNA and microRNA of Transcriptome and Proteome

DEMs, DEGs, and DEPs were analysed for an in-depth study of the infection mechanism in grass carp. In order to understand the relationship between the level of protein and gene expression, a nine-image diagram was created. In addition, an interactive network of DEMs/DEGs/DEPs was constructed and analysed using the Interacting Genes (STRING) database, and the network diagram was generated using Cytoscape3.10.2.

### 4.10. Statistical Analysis

All data represent the mean and standard deviation of three biological replicates. GraphPad Prism v.5.00 (GraphPad Software, San Diego, CA, USA) was used for comparative analysis of means. One-way ANOVA method with Tukey’s post-test was used for the analysis of DEGs and DEPs, with *p*-values ≤ 0.05 considered as the significance threshold.

## Figures and Tables

**Figure 1 ijms-25-06438-f001:**
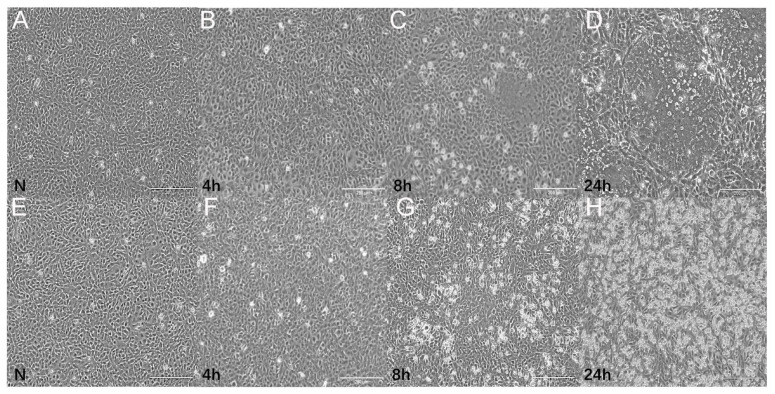
CIK cells infected with GCRV (**A**–**D**) and *Aeromonas hydrophila* (**E**–**H**) at different time points. N represents the control group, scale bar indicates 200 µm.

**Figure 2 ijms-25-06438-f002:**
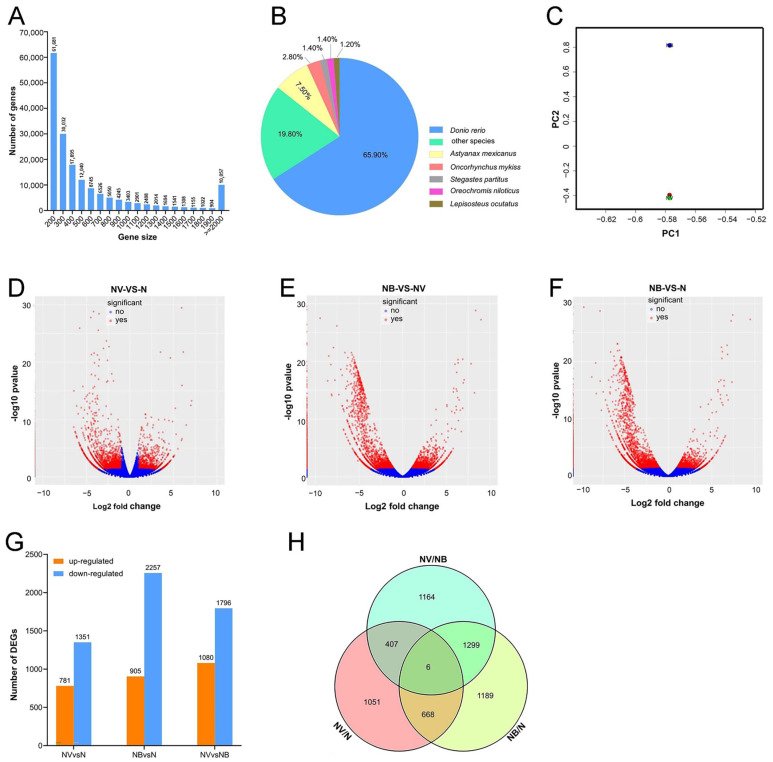
Transcriptome sequencing analysis. (**A**) Length distribution of unigenes. (**B**) Blastx analysis of unigenes from the grass carp transcriptome. Different colors represent different species; the size of the area indicates the proportion of each species. (**C**) Principal Component Analysis. PC1 represents the difference in infected samples, while PC2 represents the difference between the control group and the experimental group. The blue dot represents the bacterial group (NB), the green dot represents the viral group (NV), and the red dot represents the control group (N). (**D**–**F**) The DEGs from three treatment samples were visualised by volcano plots. The absolute values of log2 ratio ≥ 2 and *p* ≤ 0.05 (−log 10 ≥ 1.31) were performed as the threshold to assigned DEGs. (**G**) The number of up-regulated and down-regulated DEGs in each group. (**H**) The Venn diagram illustrates the overlapping situation of DEGs within each group.

**Figure 3 ijms-25-06438-f003:**
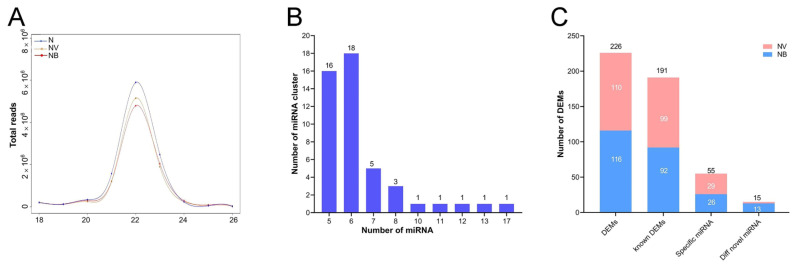
Analysis of small RNA sequencing. (**A**) Length distribution of miRNA. The red line represents the N group, the yellow line represents the NV group, and the blue line represents the NB group. (**B**) Statistics of miRNA clusters. The x-coordinate is number of miRNA clusters, and the y-coordinate is number of miRNAs. (**C**) Statistics of DEMs and specific miRNAs.

**Figure 4 ijms-25-06438-f004:**
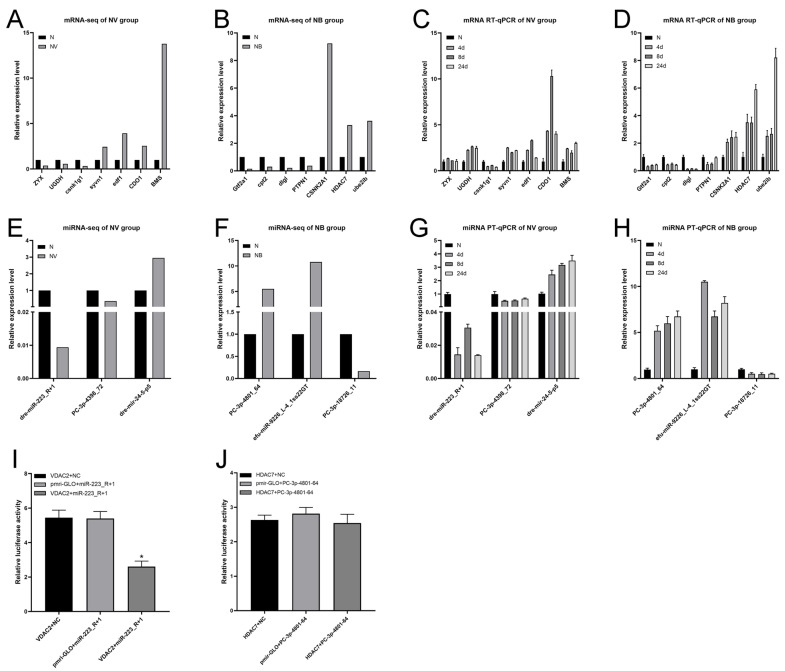
Validation of mRNA-seq/small RNA-seq by RT-qPCR. (**A**–**D**) DEGs validation of NV and NB groups. (**A**,**B**) represent transcriptome sequencing data of DEGs of NV and NB groups. (**C**,**D**) represent RT-qPCR data of DEGs of NV and NB groups. (**E**–**H**) DEMs validation of NV and NB group. (**E**,**F**) represent transcriptome sequencing data of DEMs of NV and NB groups. (**G**,**H**) represent RT-qPCR data of DEMs of NV and NB groups. (**I**,**J**) miR-223_R+1 and PC-3p-4801-64 target validation. CIK cells were transfected with miR-223_R+1 and PC-3p-4801-64, along with VDAC2 and HDAC7-3′UTR for 24 h, respectively, and the luciferase activity was determined. All data represent the mean ± S.D. of three replicates. * *p* < 0.05.

**Figure 5 ijms-25-06438-f005:**
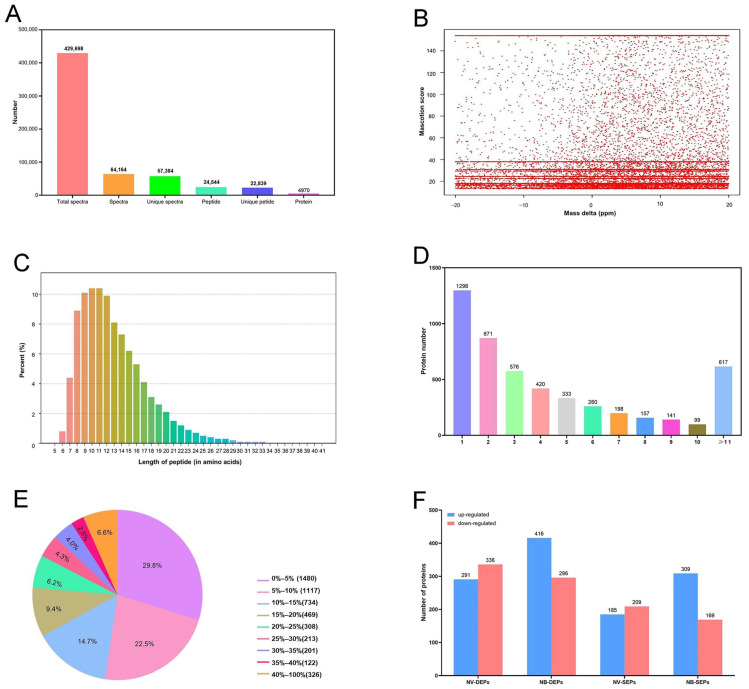
Quality control validation of mass spectrometry (MS) data. (**A**) A number of total proteins. (**B**) Mass error distribution of all identified peptides. The distribution of mass error is near 20. (**C**) Peptide length distribution. The length of most peptides is distributed between 7 and 16, which is consistent with the properties of the tryptic peptides. (**D**) Distribution of peptide numbers. Most proteins included 1–4 peptides, which is credible. (**E**) Protein coverage distribution. Protein coverage of approximately 50% peptides was more than 10%. (**F**) The number of up-regulated and down-regulated DEPs in each group.

**Figure 6 ijms-25-06438-f006:**
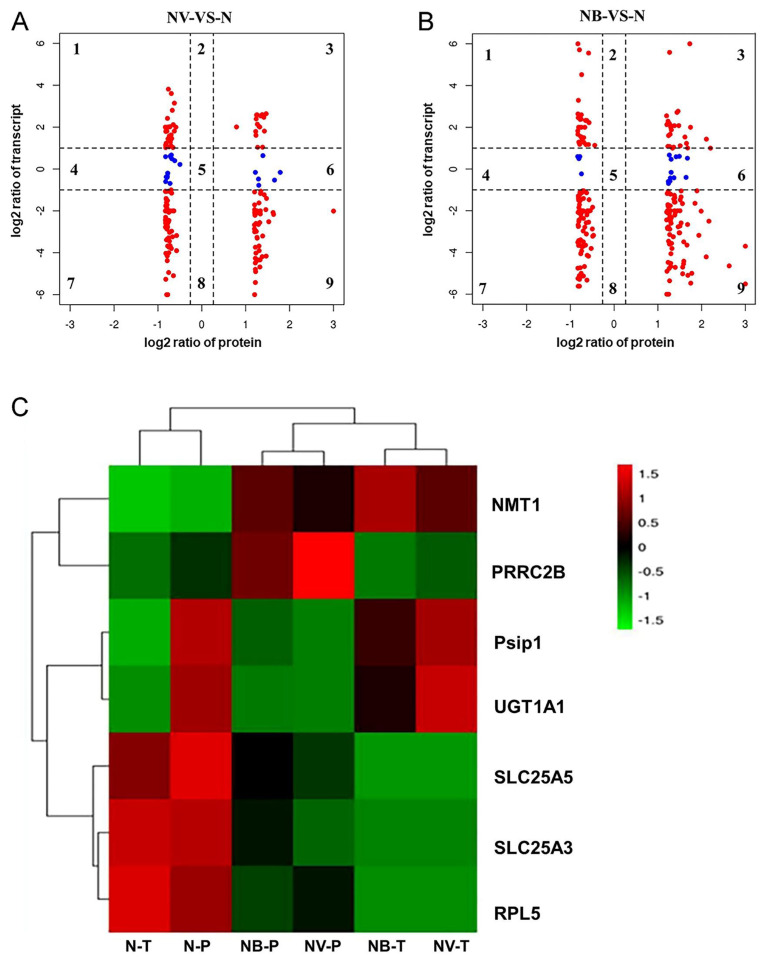
Overview interactions of miRNA–mRNA–protein. (**A**,**B**) The relationship between mRNA and protein expression levels. (**A**) NV group compared with N group. (**B**) NB group compared with N group. In the figure, it is divided into 9 quadrants. The ordinate represents the fold change of the gene expression profile, and the abscess represents the fold change of the protein expression profile. When |fold change|(FC) ≥ 2 was assigned as DEGs (log2 = 1 or −1) and fold change |(FC) ≥ 1.2 was assigned as DEPs (log2 = 0.5849, or −0.5849). Red points represent |fold change|(FC) ≥ 2, blue points represent |fold change|(FC) ≥ 1.2. (**C**) Hierarchical clustering of DEGs and DEPs common in N, NV, and NB. DEGs’ expression level of the transcriptome in different groups (N-T, NV-T, and NB-T). DEPs’ expression level of the proteome in different groups (N-P, NV-P, and NB-P).

**Figure 7 ijms-25-06438-f007:**
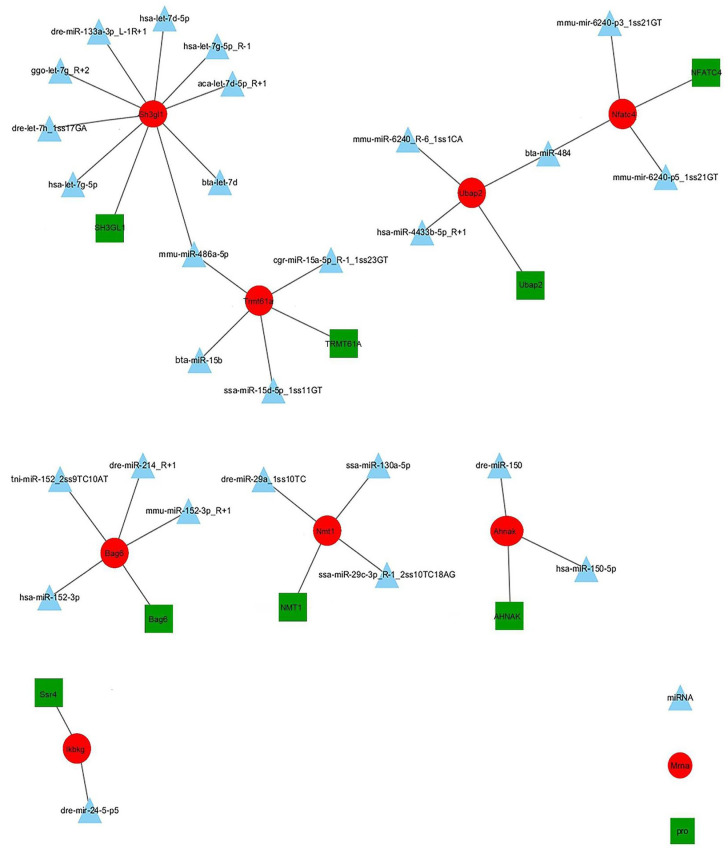
miRNA–mRNA–protein negative correlation network in NV group. Triangles represent DEMs, circles represent DEGs, and squares represent DEPs.

**Figure 8 ijms-25-06438-f008:**
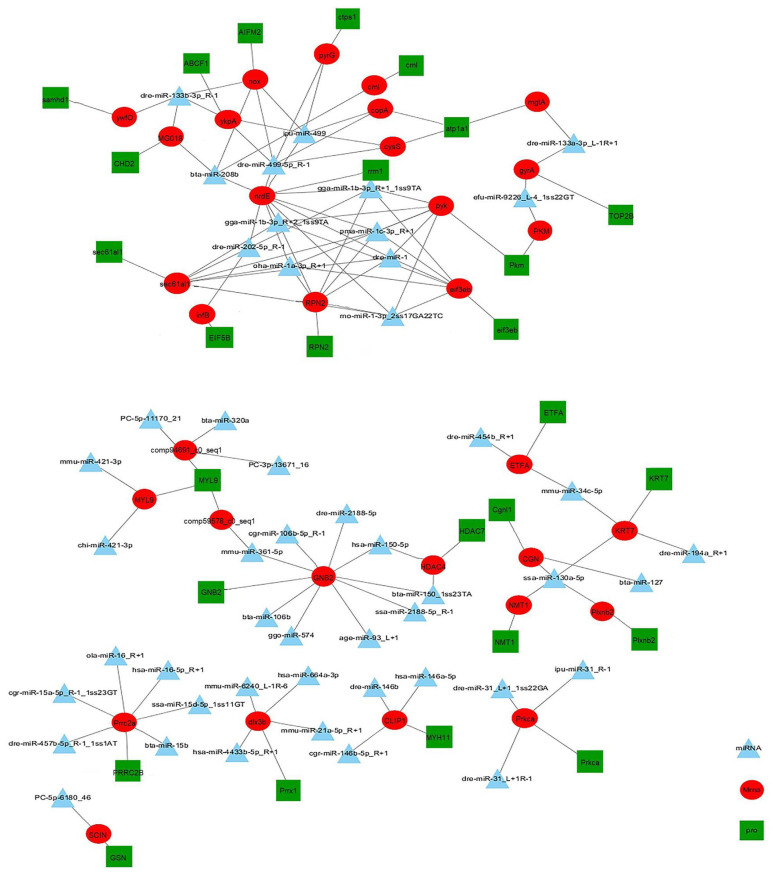
miRNA–mRNA–protein negative correlation network in NB group. Triangles represent DEMs, circles represent DEGs, and squares represent DEPs.

**Table 1 ijms-25-06438-t001:** The number of CIK cells obtained after each incubation time point in the NV and NB groups.

Sample	N	4 h	8 h	24 h
NV	1.00 × 10^6^	0.77 × 10^6^	0.43 × 10^6^	0.14 × 10^6^
NB	1.00 × 10^6^	0.80 × 10^6^	0.47 × 10^6^	0.21 × 10^6^

**Table 2 ijms-25-06438-t002:** The KEGG enrichment results of DEMs’ target genes in NV and NB groups.

Sample	KEGG Pathway	Pathway_ID	Gene Num
NV	Immune system	ko04620, ko04012, ko04062, ko04621	4
	Carbohydrate metabolism	ko00010, ko00053, ko00040	5
	Translation	ko03010	5
NB	Carbohydrate metabolism	ko00053, ko00010, ko00020, ko00030 ko00040, ko00051	14
	Membrane transport	ko02010, ko03070	9
	Transport and catabolism	ko04145, ko04142, ko04144	6
	Immune system	ko04612, ko04621	4

Note: The names corresponding to the Pathway ID are shown in Appendix A.

## Data Availability

Sequencing data generated in the study are available in the NCBI Sequence Read Archive (SRA) under BioProject accession PRJNA1118260.

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
