# Peer review of "Comparative Analysis of mRNA, microRNA of Transcriptome, and Proteomics on CIK Cells Responses to GCRV and Aeromonas hydrophila"

_ijms, 2024, doi:10.3390/ijms25126438_

Round 1

Reviewer 1 Report

Comments and Suggestions for Authors

The publication titled: „Comparative analysis of mRNA, microRNA of transcriptome and proteomics on CIK cells responses to GCRV and Aeromonas hydrophila” by Lin Y. et al. is a very good study with application of many advanced molecular analyses. In my opinion minor revisions are needed for the paper’s acceptance. I have listed some suggestions that should be included in the text of manuscript.

1.      Please insert a paragraph about in-vitro tests conducted on cell cultures in the Introduction section

2.      Please expand the sentence in lines 70-71 - description about proteomics with examples of analysis of freshwater fish

3.      Please correct the text in lines 370-373. In its current state, the text is unintelligible. Please complete the sentences with examples of analyzes on fish or other organisms.

4.      At the end of the introduction chapter, the benefits of the conducted research should be described, so that the reader, in addition to the advantages of basic research, can see the impact on the practical application of this research (you can describe what further research the obtained analytical results will be useful for).

5.      Please add the characteristics of fish from which the kidney cells were collected (e.g. age, sex, weight of fish, weight of head kidney, length of fish, time of head kidney collection, method of kidney collection, period of kidney storage).

6.      How many viable cells were obtained after each incubation time point – please add this information in line 423.

Author Response

Thank you very much for taking the time to review this manuscript. Please find the detailed responses below. We have highlighted the corrections in the re-submitted files.

Comments 1: Please insert a paragraph about in-vitro tests conducted on cell cultures in the Introduction section.

Response 1: Thank you for your valuable suggestions. We have added information in lines 51-57 of the introduction regarding the utilization of CLK cells for conducting certain immune-related in vitro experiments.

Comments 2: Please expand the sentence in lines 70-71 - description about proteomics with examples of analysis of freshwater fish.

Response 2: Thanks to your suggestion, we have added a description of the relevant proteomics as well as some proteomic analyses of freshwater fish on page 2, lines 77-82 in the text.

Comments 3: Please correct the text in lines 370-373. In its current state, the text is unintelligible. Please complete the sentences with examples of analyzes on fish or other organisms.

Response 3: Thanks to your suggestion, we have explained the corresponding content in more detail in the Discussion section of the text and added relevant examples of other organisms on page 13, lines 381-390.

Comments 4: At the end of the introduction chapter, the benefits of the conducted research should be described, so that the reader, in addition to the advantages of basic research, can see the impact on the practical application of this research (you can describe what further research the obtained analytical results will be useful for).

Response 4: Sincerely thank you for your suggestion, and we have added it accordingly: “It further provides important new information on the immune response induced by infections of viruses and bacteria in teleost fish, such as grass carp.” The change is on page 2, lines 95-96 of the text.

Comments 5: Please add the characteristics of fish from which the kidney cells were collected (e.g., age, sex, weight of fish, weight of head kidney, length of fish, time of head kidney collection, method of kidney collection, period of kidney storage).

Response 5: Thank you for pointing this out. We are so sorry that we cannot provide the characteristics of the fish from which the kidney cells were collected, because C. idellus (Ctenopharyngodon idellus) kidney (CIK) cells, Aeromonas hydrophila (AH10), and Grass carp reovirus (GCRV, JX01) were generously provided by Yang Xianle's research team at the National Aquatic Pathogens Repository, Shanghai Ocean University. This is described in Part 4.1, lines 451-455, on page 14 of the main text.

Comments 6: How many viable cells were obtained after each incubation time point – please add this information in line 423.

Response 6: Thanks to your suggestion. We have added Table 1: "The number of CIK cells obtained after each incubation time point in the NV and NB groups" on page 3, line 112, to illustrate the variations in the number of viable cells at different time points. The specific description is on lines 107-109.

Reviewer 2 Report

Comments and Suggestions for Authors

This study by Lin et al describes a systematic comparison of transcriptomic and other data from matched groups of cultured CIK cells, comparing a control group to infection with either GCRV or Aeromonas hydrophila. As both of these microbes cause hemorrhagic disease in carp, this study has the potential to serve as a starting point for mechanistic investigations into those diseases. I think the omics data in this study for the GCRV infection could be useful for that purpose, but, as the authors note, numerous studies have used similar methods (https://www.google.com/search?client=firefox-b-1-e&q=cik+cells+GCRV+infection). Because the novelty is dependent upon the comparison between the bacterial and viral infections, I think it is essential that the authors determine to what extent nutrient stress could impact the results of the NB group. The authors found several changes in metabolic pathways and 24 hours is a long time for potentially exponentially-growing bacteria. The authors may wish to calculate the MOI for the NB group and compare it to previous studies or to measure the changes in the medium after 24 hours of A. hyrophilia growth. If the CIK cells are not being strongly impacted by competition for nutrients with the bacteria, the results are likely to be more interesting to potential readers.  

Some specific comments on content and editing are included below.

Content:

Line 47:

In mammals, hematopoietic stem cells contain cells of the immune system [13, 14].

Line 77:

What is NBS? The acronym is not explained.

Figure 1: where are the images for the control (NBS) group?)

Comments on the Quality of English Language

Editing:

Line 51:

In vitro should be italicized

Lines 37-39:

once the outbreak, which would cause a large number of  grass carp deaths a, lead seriously economic and seriously threaten our country’s sustainable development of freshwater aquaculture [3-5].  (I think there’s a full stop missing here and a 2nd sentence that needs to be rephrased.

Line 85:

2.1. Viruses and bacteria change effect2.1.1. Subsubsection (is the desired subheading?)

Figure 3B: clusters, plural

Author Response

Thank you very much for taking the time to review this manuscript. Please find the detailed responses below. We have highlighted the corrections in the re-submitted files.

Comments 1: Because the novelty is dependent upon the comparison between the bacterial and viral infections, I think it is essential that the authors determine to what extent nutrient stress could impact the results of the NB group.

Response 1: Thank you for your offer. CIK cells are not significantly affected by competition with bacteria for nutrients. First, the cell culture medium contains a higher concentration of glucose along with coenzymes and cofactors necessary for cell growth, making it more conducive for cell proliferation. Secondly, research [1] has demonstrated that comparing the growth rate and proliferation characteristics of CIK cells in three different concentrations of medium revealed very similar rates of cell proliferation. Even if there are bacteria competing with the cells for the nutrients in the medium, the cells in the NB group do not die in large numbers within 24h.
[1] Zeng Lingbing, Yang Xianle, Zuo Wengong, et al. Culture of CIK cell line by using an inexpensive medium[J]. Freshwater Fisheries, 1993, (05):8-10.

Comments 2:  Line 47: In mammals, hematopoietic stem cells contain cells of the immune system [13, 14].

Response 2: We don't think this sentence clearly expresses the central idea, so we have revised it to: “In mammals, hematopoietic stem cells contain cells of the immune system [13, 14]. In fish, the kidney functions as the primary hematopoietic organ, with the channel catfish kidney (CIK) being susceptible to GCRV [15-18].” This sentence is located on page 2, lines 49-51 of the article.

Comments 3: What is NBS? The acronym is not explained.
Figure 1: where are the images for the control (NBS) group?)

Response 3: Thank you for pointing this out. We agree with this comment. Therefore, we have made the following changes: NBS was misdescribed and has been corrected to PBS, the control image is Figure 1, group N. The three group N, NB, NV descriptions are located within the article at 4.1, page 15, lines 458-460.

Comments 4:  Line 51: In vitro should be italicized

Response 4: Thank you for pointing this out. But we have removed this sentence and replaced it with something else, located on page 2, lines 51-57.

Comments 5:  Lines 37-39: once the outbreak, which would cause a large number of grass carp deaths a, lead seriously economic and seriously threaten our country’s sustainable development of freshwater aquaculture [3-5].  (I think there’s a full stop missing here and a 2nd sentence that needs to be rephrased. 

Response 5: Thank you for pointing this out. We have changed, in the first paragraph of the introduction, lines 37-41.

Comments 6:  Line 85: 2.1. Viruses and bacteria change effect2.1.1. Subsubsection (is the desired subheading?)

Response 6: Thank you for pointing this out. We agree with this comment. 2.1.1 is an unneeded subheading and has been deleted. 

Comments 7: Figure 3B: clusters, plural

Response 7: Thank you for pointing this out. We agree with this comment. The plural format in Figure 3B has been changed and is located on page 5, lines 167-170.
